# Morphological description of a novel synthetic allotetraploid(A$_1$A$_1$G$_3$G$_3$) of *Gossypium herbaceum* L.and *G.nelsonii* Fryx. suitable for disease-resistant breeding applications

Xiaomin Yin[ID]^☯, Rulin Zhan^☯, Yingdui He, Shun Song, Lixia Wang, Yu Ge, Di Chen[ID]*

Haikou Experimental Station, Chinese Academy of Tropical Agricultural Sciences, Haikou, China

☯ These authors contributed equally to this work.
* chendi209@163.com

**Data Availability Statement:** All relevant data are within the manuscript and its Supporting Information files.

## Abstract

Wild species of *Gossypium* ssp. are an important source of traits for improving commercial cotton cultivars. Previous reports show that *Gossypium herbaceum* L. and *Gossypium nelsonii* Fryx. have better disease resistance characteristics than commercial cotton varieties. However, chromosome ploidy and biological isolation make it difficult to hybridize diploid species with the tetraploid *Gossypium hirsutum* L. We developed a new allotetraploid cotton genotype (A$_1$A$_1$G$_3$G$_3$) using a process of distant hybridization within wild cotton species to create new germplasms. First of all, *G. herbaceum* and *G. nelsonii* were used for interspecific hybridization to obtain F$_1$ generation. Afterwards, apical meristems of the F$_1$ diploid cotton plants were treated with colchicine to induce chromosome doubling. The new interspecific F$_1$ hybrid and S$_1$ cotton plants originated from chromosome duplication, were tested via morphological and molecular markers and confirmed their tetraploidy through flowrometric and cytological identification. The S$_1$ tetraploid cotton plants was crossed with a TM-1 line and fertile hybrid offspring were obtained. These S$_2$ offsprings were tested for resistance to Verticillium wilt and demonstrated adequate tolerance to this fungi. The results shows that the new S$_1$ cotton line could be used as parental material for hybridization with *G. hirsutum* to produce pathogen-resistant cotton hybrids. This new S$_1$ allotetraploid genotype will contributes to the enrichment of *Gossypium* germplasm resources and is expected to be valuable in polyploidy evolutionary studies.

## Introduction

The *Gossypium* genus includes 54 species, of which 50 are wild cotton species and four are cultivated cotton species made up of two diploids and two tetraploids [1, 2]. Approximately 90% of commercially produced cotton is cultivated from *G. hirsutum* L. However, as the high-yielding quality of cultivated species continuously increases, abiotic and biotic resistance usually does not increase and can even be weakened [3, 4]. To increase the resistance in *G. hirsutum*

**Funding:** This research work was financially supported by the National Natural Science Foundation of China (No. 31701478) and Central Public-interest Scientific institution Basal Research Fund for Chinese Academy of Tropical Agricultural Sciences(No.1630092017004).

**Competing interests:** The authors have declared that no competing interests exist.

L., the existing genetic basis of cultivated cotton should be expanded to adapt to biotic and abiotic stresses. It is generally believed that wild cotton species contain a large treasure trove of traits that could be used to improve cultivated species [5]. The wild cotton species are widely distributed in various ecological conditions in the Americas, Oceania, Africa, and Asia. These species have undergone long-term natural selection, and are resistant to many types of adverse factors, such as disease, insects, drought, heat, and salinity. At the same time, there are many mutant varieties, all of which contribute to cotton's rich genetic foundation [6, 7].

*G. herbaceum* L. is a diploid cotton species cultivated in Western China and Russia. This species is highly resistant to drought, cotton leaf curl virus and sap-feeding insects such as leaf-hoppers, whiteflies, thrips, and aphids [8, 9]. *G. nelsonii* Fryx. is a wild diploid cotton species that originated in Australia and has many economically valuable characteristics, such as delayed pigment gland formation. This is beneficial for producing seeds with low gossypol levels. In addition, it is resistant to blight, Verticillium wilt, aphids, and mites; and is tolerant to abiotic stress conditions such as high temperatures and drought. Furthermore, its brown, straight, extensible, and high-strength fibers are valuable agronomical traits. If these characteristics can be transferred to tetraploid *G. hirsutum* L. ($2n = 4x = 52$, AADD), then the resulting species will be extremely valuable.

Owing to biological isolation barriers, it is difficult to directly hybridize wild diploid cotton species with tetraploid *G. hirsutum*. We are proposing a new pathway for transferring genes from two diploid cotton species into tetraploid *G. hirsutum*. Specifically, two different diploid species undergo interspecific hybridization so that they can recognize and overcome their biological segregations to form the diploid hybrid $F_1$. Following that, artificial ploidy doubling of the apical regions in $F_1$ hybrids is carried out to obtain a tetraploid cotton species that has the same chromosome number as *G. hirsutum* L. The new tetraploid cotton is then hybridized with the tetraploid *G. hirsutum* to achieve efficient gene transfer. This process shortens the time needed to introduce essential genes from two species and is currently the most direct and easiest method that has the potential to be utilized with wild cotton. In the last few decades, studies on interspecific hybrids have been reported, including *G. arboretum* × *G. raimondii* [10], *G. hirsutum* × *G. sturtianum* [11], *G. arboreum* × *G. bickii* [12], *G. hirsutum* × *G. klotzschianum* [13], *G. hirsutum* × *G. trilobum* [14], *G. arboreum* × *G. anomalum* [15], *G. hirsutum* × *G. anomalum* [16], *G. herbaceum* × *G. australe* [17], *G.hirsutum* × *G.australe* [18], *G. hirsutum* × *G. arboretum* [19], and *G. capitis-viridis* × $(G.hirsutum × G.australe)^2$ [20]. To the best of our knowledge, there has been no report of using *G. herbaceum* and *G. nelsonii* to successfully synthesize an allotetraploid ($A_1A_1G_3G_3$).

In this work, we applied homoploid hybridization to provide a pathway to simultaneously transfer genes from these two diploid cotton species into *G. hirsutum* using *G. herbaceum* as the female parent and *G. nelsonii* as the male parent to successfully generate the interspecific hybrid $F_1$ ($2n = A_1G_3 = 26$). Colchicine treatments were used to double the chromosome number and form a new tetraploid cotton genotype (code: $S_1$) ($2n = 4x = 52$, $A_1A_1G_3G_3$). The birth of the new $S_1$ cotton tetraploid not only enriches the germplasm of cotton but also provides a valuable intermediate material for cotton hybridization and a new tool for breeding. This is a successful example of using wild cotton species for distant hybridization and has important significance for promoting scientific research on cotton genetics.

## Materials and methods

### Plant material

The diploid *G. herbaceum* cotton species that has been cultivated in China is also known as Hongxingcaomian ($2n = 2x = 26$, $A_1A_1$), and is a line that has not been subjected to

inbreeding. *G. nelsonii* is a diploid cotton species from Australia ($2n = 2x = 26$, $G_3G_3$), and the two *G. hirsutum* lines, cv. *Jimian11* (a commercial variety from China) and the control variety TM-1 (a prototype for molecular mapping in cotton) were used in our experiments. The lines mentioned above were obtained from the China National Wild Cotton Nursery in Sanya, China. The *Verticillium dahliae* strain Dahlia Vd080, used for disease resistance identification, was provided by the Institute of Cotton Research, Chinese Academy of Agricultural Sciences.

## Interspecific hybridization

Interspecific hybridization of the cotton was carried out in Sanya (18°25′34″N, 109°28′25″E), Hainan, China in 2015 [21]. When the two parental species flowered at the same time, *G. herbaceum* plants were selected as the female parents. One day before flowering, all stamens in the flowers from the female parents were completely removed. Wax tubes were used to cover the stigmas and non-woven bags were used to cover all of the floral buds. The afternoon before flowering, cotton thread was used to tie the apices of the petals in the flowers of the male parents (*G. nelsonii*), and non-woven bags were used to cover the floral buds entirely. From 9 am to 12 pm on the day of flowering, pollen from the male parents was used to pollinate the stigmas of the female parents for hybridization. After pollination was completed, wax tubes were used to cover the stigmas, non-woven bags were used to cover the entire floral buds, and labels were hung. After the cotton bolls had matured, the seeds were removed and sown in nutrient pots containing sterile soil. After one or two true leaves had sprouted, the plants were transplanted into fields and normal fertilizer and water management was carried out. The morphological characteristics of the plants were observed, including the type of plant, stems, leaves, flowers and bolls. All these observations for each plant part at different biological stages were photographed using a digital camera, and the photographs were edited by using Photoshop software.

## SSR molecular marker analysis in $F_1$ hybrids

We randomly selected 568 pairs of SSRs based on our cotton genetic map [22] to screen for parental polymorphisms. The polymorphic primers obtained were used to validate the authenticity of the interspecific $F_1$ hybrids. These SSR primer sequences can be found at http://www.cottonmarker.org. A programmable gradient thermal cycler (Bio-Rad T100) was used for SSR-PCR amplification [23]. Electrophoresis and silver staining of the PCR products were carried out according to previously reported methods [24]. The bands generated by electrophoresis were photographed and edited by Photoshop software.

## Chromosome duplication in the $F_1$ interspecific hybrid

In 2016–2018, the $F_1$ hybrids that were validated by the SSR molecular markers were grafted. The temperature was kept at 25°C and the apical growth points of the $F_1$ hybrids were immersed in 0.30% (w/v) colchicine for 6 h. After immersion, they were placed in pots in the greenhouse that were not exposed to light and incubated until new growth points or floral buds sprouted. When the length of the new meristematic apices reached 5 cm, they were removed and grafted for storage and further validation of the chromosome doubling results [20]. Newly sprouted floral buds could grow until the afternoon before flowering. Then, a cotton thread was used to tie the tip of the petal for self-pollination and non-woven bags were used to cover the entire floral bud to prevent outcrossing. After the cotton bolls matured and underwent dehiscence, the seeds were removed and sown in nutrient pots. After one or two true leaves sprouted, the plants were transplanted into fields. The morphological characteristics of the plants were observed, including the plant shape, stems, leaves, flowers, and bolls.

The resulting tetraploid plant was named $S_1$, and chromosome counting and flowcytometric analysis were performed for validation. The morphological characteristics of each plant part were photographed using a camera, and the photographs were edited by using Photoshop software.

## Ploidy analysis

In order to detect whether the $S_1$ chromosome had been doubled, flow cytometry was used to examine the differences in nuclear DNA content between $F_1$ hybrid and $S_1$ cotton plants using also the diploid *G. herbaceum* and the tetraploid *G. hirsutum* (cv TM-1) as controls. The operational steps were divided into preparation of the sample cell suspensions, specific staining of DNA, and loading. Specific methods were carried out according to the description of Galbraith) [25]. A Sysmex-Partec CyFlow Cube8 flow cytometer was used to determine the nuclear DNA content. CellQuest Pro software was used to analyze the results. The experimental data are displayed in a single-parameter histogram. The ordinate represents the number of nuclei (counted), and the abscissa represents the relative intensity of the fluorescent channel.

## Karyotype analysis

Microscopic observations on the number of root tip chromosomes in the $F_1$ hybrid and its resultant tetraploid $S_1$ progenies were carried out to observe the differences in the chromosome number between the two genotypes, and *G. herbaceum* and *G. nelsonii* which were used as controls. Finally, karyotype analysis of the $F_1$ hybrid was performed for determination of chromosome doubling. Root tip fixation, preservation, dissection, squash preparation, and microscopy were carried out according to previously reported methods [26]. The editing of all images was done using the Photoshop software.

## Validation of disease resistance in the tetraploid $S_2$

The $S_2$ offsprings were chosen as the target of disease resistance identification. The tetraploid $S_2$ seedlings, growing in nutrient pots, were inoculated indoors with the *Verticillium dahliae* strain (code: Dahlia Vd080 strain) for preliminary determination of disease resistance. The strain cv. *Jimian11* was used as an infection control group; *G. herbaceum* and *G. nelsonii* were used as disease resistance controls. In the experiment, there were two treatments (fungal suspension or sterile water for inoculation) performed in triplicate. Each replicate of the experiment contained 540 plants.

Inoculations of the seedlings in the nutrient pots were carried out according to a previously reported method [27]. Seeds were sown in cylindrical paper pots (height 8 cm and diameter 5 cm) with no bottom and filled with sterile medium. When the first true leaf had grown until it was level, $10^7$ conidia/ml of suspension was directly injected into the 8-cm-dimeter plastic tray. Only sterile water was used for inoculation in the control group. The paper pot and seedlings were placed upright in the trays so that the root systems adsorbed the suspension liquid. Examination of the disease symptoms was carried out according to previously reported methods [19]. Changes in leaf color and degree of wilting were used as a basis for preliminary observations of disease resistance in the experimental materials.

## Hybridization between the $S_1$ interspecific hybrid and *G. hirsutum* (TM-1 cultivar)

In the winter of 2017, hybridization was carried out in Sanya, Hainan province, China, using the tetraploid $S_1$ as the male parent (♂) and the TM-1 line (♀) as the female parent. The

hybridization method used was similar to the previously described for *G. herbaceum* (♀) and *G. nelsonii* (♂) [21], and the boll-forming rate and effective boll-forming rate of hybrids were determined. After the hybridized cotton bolls matured and underwent dehiscence, seeds were removed and sowed in nutrient pots containing sterile soil. After one or two true leaves sprouted, the plants were transplanted into fields. The morphological characteristics of the plants were observed, including plant shape, stems, leaves, flowers and bolls. Based on the morphological characteristics, the authenticity of the hybrids was preliminarily assessed and the fertility of the tetraploid $S_1$ hybrids was examined.

### Image acquisition and editing

A Canon EOS 70D digital camera with an RF 35 mm F1.8 Macro Lens was used to photograph the morphology of plant parts and electrophoresis bands of the PCR products. The images were edited with Adobe Photoshop CS5 software.

## Results

### $F_1$ hybrid generation and trait observations

In 2016, hybridization was conducted using *G. herbaceum* as the female parent and *G. nelsonii* as the male parent. A total of 235 flowers were pollinated, 205 of which formed bolls. However, most bolls did not form seeds and only three effective bolls were finally obtained; each containing one seed. In 2017, the three seeds were sown in a pot and all germinated after 5 days. After the first true leaves were level, the seedlings were transplanted to the field. Floral buds appeared after 43–45 days and flowering occurred after 68–70 days. In Sanya, China, the three $F_1$ plants showed exuberant growth, were tall, produced large numbers of flowers throughout the year, and could produce pollen. However, the pollen produced was non-viable and did not produce bolls. The samples have been preserved, but are still infertile. This observation underline the difficulty in obtaining $F_1$ hybrids from interspecific hybridizations between *G. herbaceum* L. and *G. nelsonii* Fryx.

The morphological traits of the $F_1$ plants showed classical hybrid characteristics (Fig 1) according to the 14–item investigation data analysis (Table 1). Morphological comparison with parental plants showed that two items were greater than the parents: plant height (Fig 1A–1C) and corolla color (Fig 1E–1G). Five items were intermediate between the parents: leaf shape and color, number of lobes, bract shape, calyx teeth shape, and number of bract teeth (Fig 1E–1G). Two items were similar to the female parent, anther and pollen color (Fig 1E and 1G). Five items were similar to the male parent: stem and leaf trichomes, flowering habit, bract extension, flower opening, and extrafloral nectary (Fig 1F and 1G).

SSR molecular marker determination was carried out and six pairs of parental polymorphic primers (Table 2) were obtained from the screening. These six pairs of primers could be used to amplify the complementary bands from the parents in the $F_1$ hybrids (Fig 2A), of which two primer pairs (NAU1052 and NAU3093) amplified not only the bands that are characteristic of the parents but also determined the characteristic bands of the hybrid (Figs 2A/iii and 3A/vi) showing the hybridized stature of $F_1$ hybrids. Flow cytometry showed that the peak value of the control sample of *G. herbaceum* was approximately 200 channels (Fig 2B); the peak value of TM-1 was close to 400 channels (Fig 2C); and the peak value of the $F_1$ hybrid was close to 200 channels (Fig 2D), indicating that the $F_1$ hybrid was a diploid. Root tip staining and microscopy observations showed that the root tip chromosome number of both the *G. herbaceum* and *G. nelsonii* control samples (Fig 2E and 2F), and the $F_1$ hybrid (Fig 2G) was also 26 pairs, indicating that the $F_1$ hybrid was a diploid.

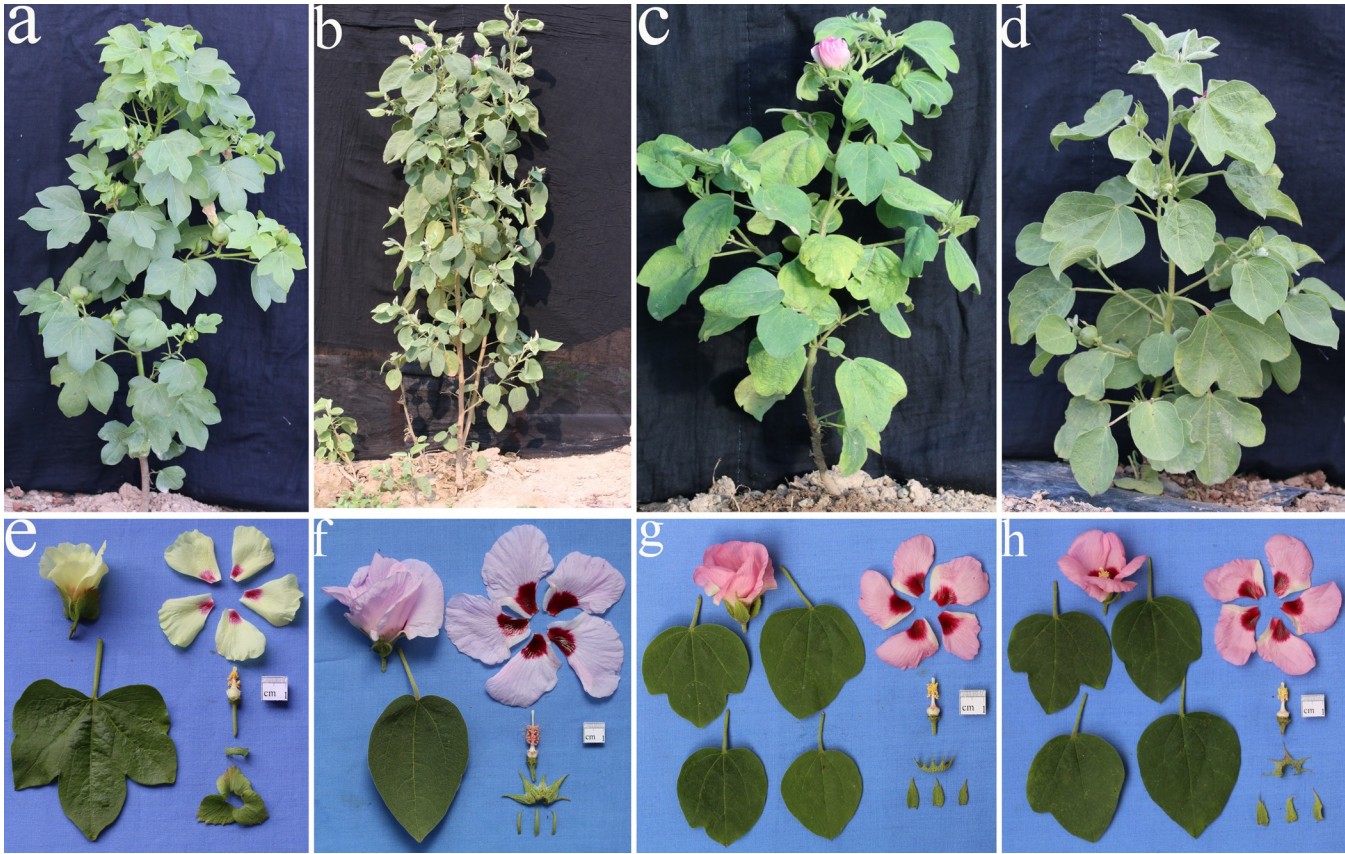

**Fig 1. Pictures of plants, leaves, flowers, corollas, basal patches, filaments, anthers, pollen, calyxes, and bracts.** *G. herbaceum* (a, e), *G. nelsonii* (b, f), the $F_1$ (c, g), the $S_1$ (d, h).

## Tetraploid $S_1$ generation and trait observations

After the growth of apical meristems of $F_1$ hybrid underwent chromosome doubling, the first large red flower grew in the apex of the plant after 27 days (S1A Fig). The flower was full of pollen and a boll containing four seeds was formed through self-pollination (S1B and S1C Fig). In 2018, the four seeds were sown in a pot and all germinated after 5 days. After the first true leaves were level, the seedlings were transplanted to the field. Floral buds appeared after 45–50 days and flowering occurred after 70–75 days. Boll opening occurred after 107–115 days. In Sanya, the four $S_1$ plants grew normally, with medium plant height and flowering that occurred throughout the year. The pollen formed was viable and many bolls were formed. The samples that were preserved were fertile. Chromosome doubling of the $F_1$ hybrids restored their fertility. We hypothesized that the four $S_1$ plants were all tetraploid hybrids and were assigned the code $S_1$ (2n = 4x = 52, $A_1A_1G_3G_3$). The differences in the morphological characteristics between the four $S_1$ plants were not large. Most plants developed large red flowers, with radiating purple patches, and significant separation was not observed. The leaves were large and tough, and the plants were tall and shrub-like (Fig 1D). The plants produced many flowers, released pollen, and produced fruit normally. The fruits formed were small bolls containing small seeds, and short, brown, high-strength fibers (Fig 3F). In particular, the $S_1$ plants retained some characteristics of *G. nelsonii* Fryx. For example, at low temperatures, the buds on the lower branches were extremely small (0.50 cm length), milky white in color, did not open until wilting, and had closed pollination. In contrast, flowers in the middle and top branches were large (4.40 cm length), opened normally, and had

**Table 1. Comparison of morphological traits between *G. herbaceum* L. × *G. nelsonii* Fryx. $F_1$ hybrid and its parental genotypes.**

| Character | *G. herbaceum* L. | *G. nelsonii* Fryx. | $F_1$ |
|---|---|---|---|
| Plant height (cm) | 80 | 200 | 300 |
| Stem/branch characteristics | Less trichomes | More trichomes | More trichomes |
| Shape and color of leaf | Broad-leaved, Green | Ovoid, gray green | Broad-leaved, ovoid, subcircular, Green |
| No. of lobes | 3–5 | 0–1 | 0–3 |
| Bract size (length × width, cm) | 1.1×1.3 | 1.2×0.2 | 1.1×0.5 |
| Bract shape, No. of teeth | Heart-shaped, 4–6 | Lanceolate, 1 | Sword-shaped, 1–3 |
| Bract jointing | Joint | No joint | No joint |
| Floral nectary | Intra-floral | Extrafloral | Extrafloral |
| Calyx | Cup-shaped | Wavy | Serration |
| Bract tooth | 5–9 | 1 | 4–6 |
| Corolla shape and color Corolla length (cm) | Yellow trumpet, around 2.1 | Light pink, funnel-shaped, around 3.5 | Pink trumpet, around 2.5 |
| Color and size of spot | Rose | Purple | Red |
| Filaments colors | Pink interspersed with milky white | Rose | Light pink |
| Anther colors | Yellow | Rose | Light yellow |
| Pollen colors | Yellow | Milky white | Yellow |
| Stigma colors | Milky white | Light green | Milky white |
| Boll | Conical | Oval | / |
| Fiber | Spiral | Upright | / |

open pollination. The red flowers contained large purple patches, the bracts were opened, and the plants contained a lot of glands, but not the seed embryo.

The most important differences in morphological characteristics between $S_1$ and $F_1$ plants, concerned to the size of leaves and viability of pollen. In general, $S_1$ plants are shorter than $F_1$ ones with larger and thicker leaves (Tables 1 and 3). Because of their viable pollen, formed bolls and seeds, a mean number of 50 bolls per plant (S2A Fig) were produced during 4 months of growth in the field. The samples were still fertile after preservation. Fertility was restored after the diploid $F_1$ hybrids underwent chromosome doubling. However, other $S_1$ traits were not different from the $F_1$ plants, including leaf shape and color, number of lobes, bract shape, calyx teeth shape, number of bract teeth, anther and pollen color, stem and leaf trichomes, bract extension, and extrafloral nectary (Fig 1G and 1H).

We determined that the $S_1$ plants were similar to the $F_1$ plants as their morphological traits showed classical hybrid characteristics (Fig 1). According to the 25-item investigation data analysis (Table 3), 10 items(Fig 1E–1H) were intermediate between the parents including leaf shape and color, number of lobes, bract shape, calyx teeth shape, number of bract teeth, corolla size, number of anthers, boll diameter (Fig 3A–3C), fiber color, and fiber length. Two items were greater than the parents, including the number of boll loculi and fiber tensile strength

**Table 2. SSR polymorphism primer sequences.**

| Primer number | Forward sequence (5′-3′) | Reverse sequence (5′-3′) |
|---|---|---|
| NAU1157 | GAGTTTGGTTCTGGGTTGAG | GATCCTTTTCATCTCCTCCA |
| NAU1355 | ATCTGTTTACGCCACTCTCC | CCAGCCTTTGACATTTTTCT |
| NAU1052 | CGCAGATAAAGGATGGATTT | AGAGCTGGAGGACATAACAAA |
| JESPR156 | GCCTTCAATCAATTCATACG | GAAGGAGAAAGCAACGAATTAG |
| BNL1679 | AATTGAGTGATACTAGCATTTCAGC | AAAGGGATTTGCTGGCAGTA |
| NAU3093 | GTCTTGAACCGGAACTTGAT | TCCTGTTGAACACCAAAGTG |

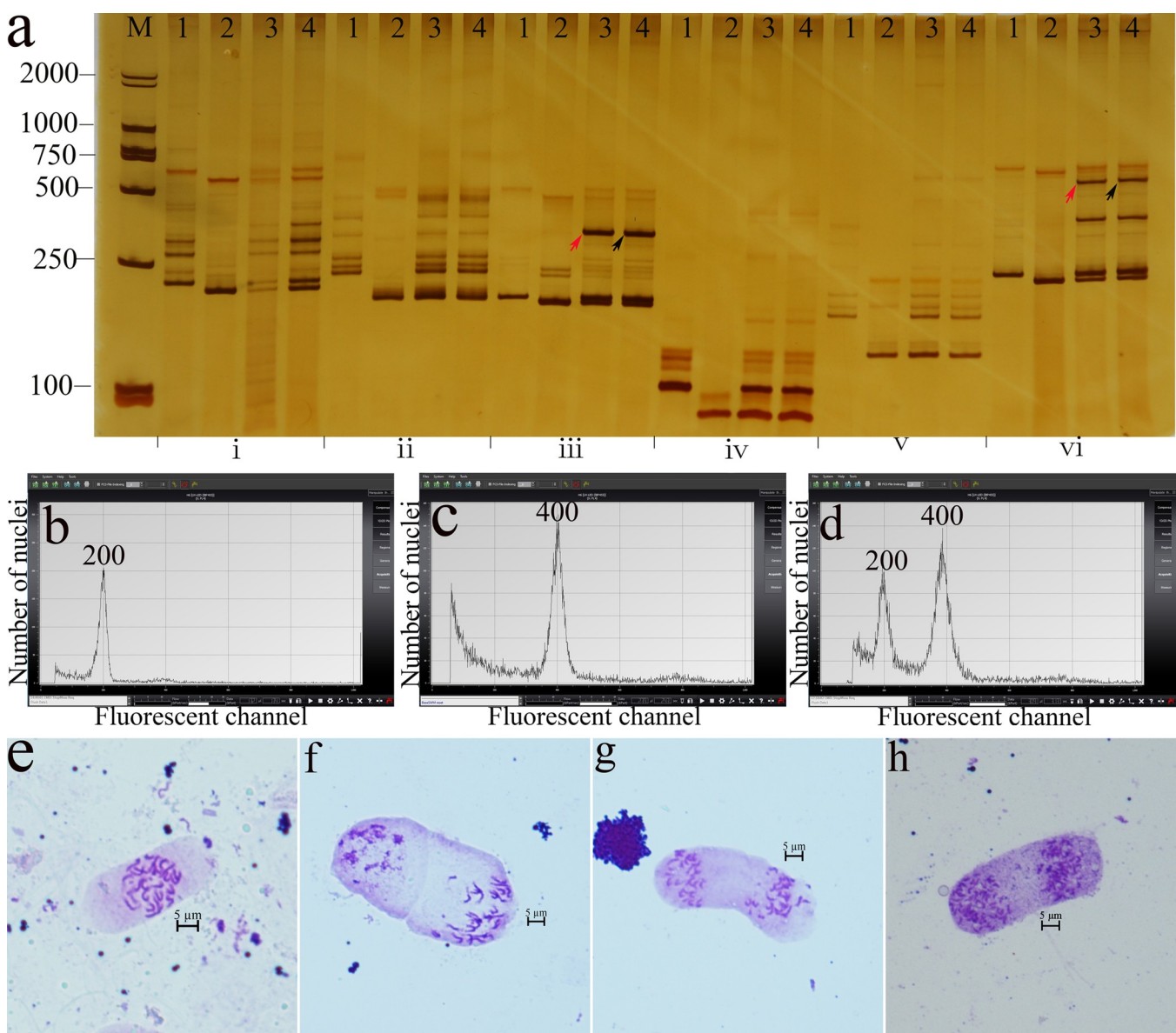

**Fig 2. Validation of F₁ and S₁ by detecting DNA polymorphism, ploidy analysis, and the number of chromosomes.** (a) Microsatellite loci identified by SSR primers including (i) NAU1157, (ii) NAU1355, (iii) NAU1052, (iv)J ESPR156, (v) BNL1679, and (vi) NAU3093. The novel bands produced in F₁ and S₁ are indicated by red and blue arrows (1, *G. herbaceum*; 2, *G. nelsonii*; 3, the F₁; 4, the S₁). (b) *G. herbaceum* at 200 channels. (c) TM-1 at 400 channels, respectively. (d) F₁ and S₁ at 200 channels and 400 channels respectively. (e) 26 of *G. herbaceum*. (f) 26 of *G. nelsonii*. (g) 26 of the F₁. (h) 52 of the S₁. The root tip cells (red arrows) and the chromosomes (black arrows) stained with Carbol fuchsin solution under 40× microscope.

(Fig 3D–3F). Two items were similar to the female parent, including anther and pollen color (Fig 1E and 1H), and six items were similar to the male parent including stem and leaf trichomes (Fig 1B and 1D), flowering habit, bract extension, bract size (Fig 1F and 1H), flower opening, and extrafloral nectary. In addition, maternal traits expressed as dominant for the anthers, stamens, and stigmata's characteristics (Fig 1E and 1H), whereas paternal traits were expressed in the stem and leaf trichomes, boll shape, and trichomes (Fig 3E and 3F).

SSR molecular marker determination showed that the six pairs of parental polymorphic primers(Fig 2A) amplified the complementary bands of the parents, as well as the characteristic

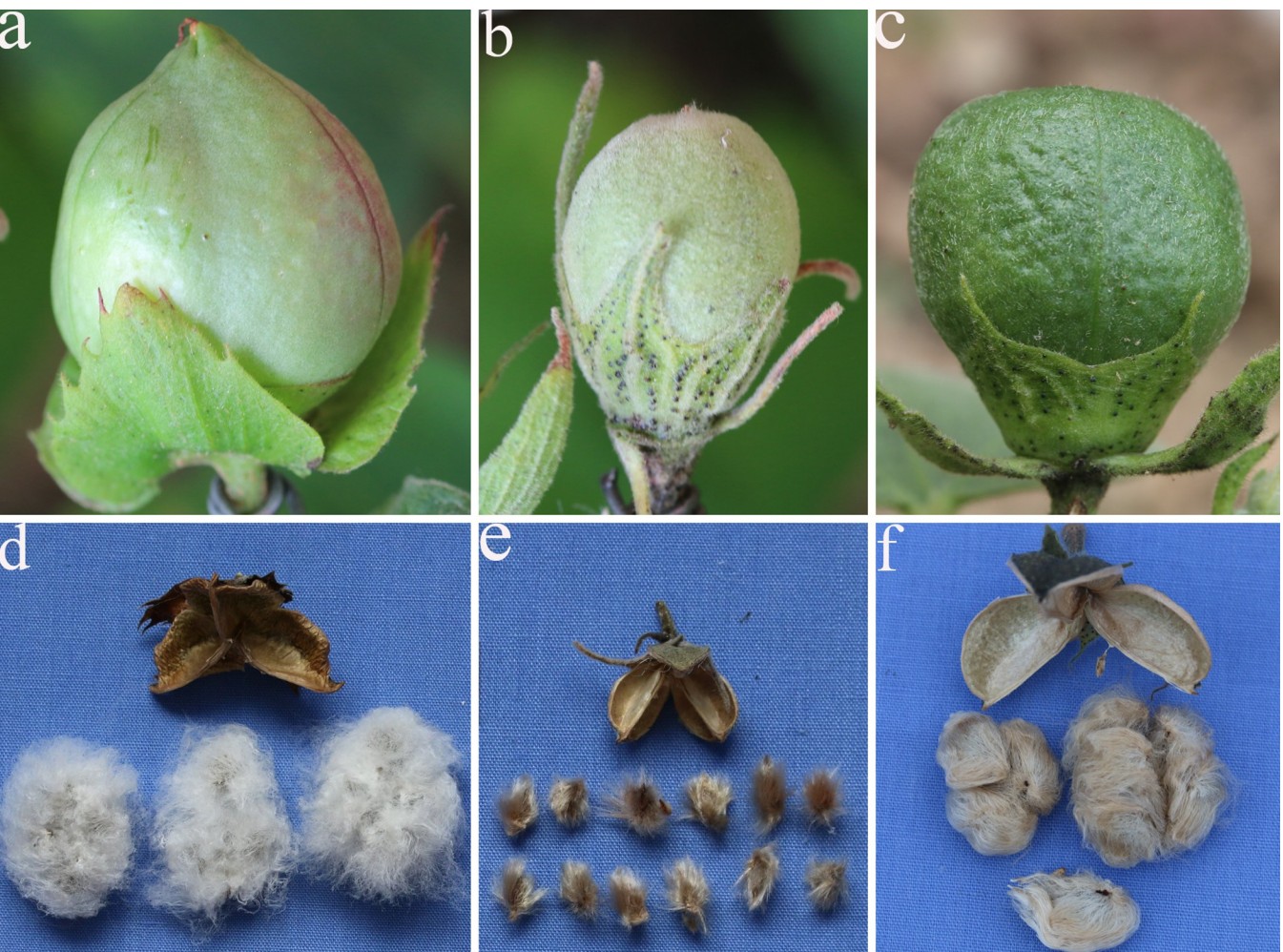

**Fig 3. Pictures of cotton bolls and fibers.** (a) *G. herbaceum* had conical, smooth green and red bolls. (b) *G. nelsonii* had oval, green cotton bolls. (c) Green pear-shaped boll of $S_1$. (d) *G. herbaceum* had 3-chambered bolls with gray-white spiral cotton fibers. (e) *G. nelsonii* had 3-chambered bolls with dark brown upright fibers. (f) The $S_1$ had 3-chambered bolls with brown fibers.

bands of the hybrid itself (Figs 2A/iii and 3A/vi). This indicated that $S_1$ is also a true hybrid. At the same time, this shows that the external genes were not introduced during the chromosome doubling of the $F_1$ to obtain the $S_1$. Flow cytometry showed that the peak value of the control sample of *G. herbaceum* was close to 200 channels (Fig 2B), the peak value of the TM-1 was close to 400 channels (Fig 2C), and the peak value of the $S_1$ hybrid was close to 400 channels (Fig 2D), showing that the $S_1$ hybrid was a tetraploid. Root tip staining and microscopy observations showed that the root tip chromosome number of the *G. herbaceum* and *G. nelsonii* control samples was 26 pairs (Fig 2E and 2F), whereas the root tip chromosome number of the $S_1$ hybrid was 52 pairs (Fig 2H). These measurements confirmed that $S_1$ hybrid was tetraploid.

## Examination of tetraploid $S_2$ segregation

The $S_2$ plants were completely different from the $S_1$ plants, and showed abnormal and intense segregation. Due to the complex genetic basis of interspecific hybrids, the segregation of $F_2$ populations was diverse, from male parent to female parent types, including a series of intermediate types, as well as new types of plants that are not available in parents. In the $F_2$

Table 3. Comparison between $S_1$ and $S_2$ (*G. herbaceum* × *G. nelsonii*) generations based on morphological and fertility traits.

| Generation | Plant height (cm) | Colors of leaf | Leaf size (length×width, cm) | No. of lobes | No. of nectaries on the back of leaf blade | Corolla color | Corolla length (cm) | Color and size of spot | Anther and Filaments colors | Bract size (length×width, cm) | Bract shape, No. of teeth (n) | Flower nectary | Boll shape, No. of boll loculi | Boll size (length×width, cm) | Sterility |
|---|---|---|---|---|---|---|---|---|---|---|---|---|---|---|---|
| $S_1$ | 150.00 | Dark green | 5.10×5.40 | 0–3 | 1 on primary vein | Pink, | Around 2.60 | Red | Yellow, Light pink | 1.20×0.60 | Sword-shaped, 1–3 | Extrafloral | Pear-shaped, 4 | Generally, 3.00×2.50 | Many bolls formed |
| $S_2$ | 75.00–170.00 | Light green, Dark green | 5.70×6.20–10.90×13.60 | 0–3 | 2, 3 at the primary vein and lateral vein | Pink, Light pink, Milky white, Red flower with white border | 2.40–4.40 | Red, Purple, Rose, None | Yellow, Light pink, Pink interspersed with milky white | 1.30×0.70–1.60×1.10 | Triangle, Dentate, Ensiform, 2–4 | None, extrafloral, intra-floral | Pear-shaped, sunken navel, oval, irregular round boll, 3–4 | 2.70×2.30–3.20×2.60 | Only 32 out of 145 plants had 1–7 bolls while no bolls were found in other plants |

generation population, 10 representative plants were selected as observation objects, and morphological indexes were compared in the $S_2$ descendants of $S_1$ with their parental plants (Table 4). For example, the shortest plant (75.00 cm) was shorter than the female parent *G. herbaceum* L., whereas most shrubs were tall and thick with the tallest plant being 170.00 cm. Corolla color began to segregate as plants came to have not only red corollas, but also pink, red with white borders, and milky white flowers (Fig 4A–4H). However, yellow flowers were not observed on the female parent (Fig 1E). The leaves were generally large, with the longest leaf being 9.90 cm and the widest leaves being 11.00 cm. Nectaries at the back of the leaves were also unique, as they were different from normal leaves that had one nectary at the base of the primary vein, but both primary and lateral veins contained nectaries. Some leaves had two, whereas other leaves had three nectaries (Table 3), which are traits rarely observed in the *Gossypium* genus.

Observations also showed that the $S_2$ morphology was associated with fertility. Plants that resembled a pagoda like the female parent produced more bolls and seeds, with 5–10 seeds per boll. In contrast, plants that resembled the male parent were inclined to have a higher ratio of sterility and few fertile plants that produced fewer bolls, with each boll containing only a few seeds (1–5).

Fertility was also found to be highly segregated, with some weaker plants growing slowly and flowering later with fewer flowers. They were still normally able to disperse pollen and fruit, even though the fruits were small and not upright. Other plants grew more readily, flowering earlier with more flowers and fruits, and the fruits were large and upright. Most fibers produced were brown, but some bolls contained fibers that were both brown and light green (S2B and S2D Fig).

The above results indicate that the $S_2$ generation underwent intense segregation in both morphological characteristics and fertility. This phenomenon exists in varying degrees in other interspecific hybrids [28] and is common in segregated generations [7]. However, it was particularly prominent in the $S_2$ generation of this hybrid. Table 3 compares the observation data of the $S_2$ and $S_1$ plants. It should be mentioned that the $S_2$ was used as an experimental material because it was a population with genetic segregation and with a rich genetic diversity.

**Table 4. Comparison of quantifiable parameters in the $S_2$ descendants of $S_1$ with their parental plants.**

| Plants | Plant height (cm) | Internode of stem (cm) | Vegetative branches | Fruiting branches | Leaf size (length×width, cm) | Bract size (length×width, cm) | Boll size (length×width, cm) | Corolla length (cm) | Stigma length (cm) | Length of fiber (mm) |
|---|---|---|---|---|---|---|---|---|---|---|
| $S_2$-1 | 156.68 | 5.18 | 5 | 13 | 8.26×9.76 | 1.58×1.10 | 3.11×2.56 | 4.35 | 1.14 | 12.86 |
| $S_2$-2 | 170.00 | 5.21 | 7 | 15 | 9.23×10.81 | 1.60×1.10 | 3.26×2.65 | 4.46 | 1.15 | 11.12 |
| $S_2$-3 | 146.23 | 5.06 | 5 | 14 | 9.51×10.72 | 1.50×0.98 | 2.89×2.67 | 4.14 | 1.22 | 10.46 |
| $S_2$-4 | 138.17 | 4.89 | 6 | 15 | 8.62×9.92 | 1.30×0.89 | 3.05×2.38 | 3.95 | 1.21 | 12.98 |
| $S_2$-5 | 129.28 | 5.07 | 5 | 12 | 9.81×10.61 | 1.25×0.76 | 2.97×2.56 | 4.28 | 1.36 | 10.56 |
| $S_2$-6 | 89.86 | 4.86 | 4 | 9 | 6.33×7.50 | 1.38×0.72 | 2.84×2.80 | 2.51 | 1.28 | 9.78 |
| $S_2$-7 | 70.00 | 3.16 | 4 | 7 | 5.71×6.20 | 1.3×0.72 | 2.78×2.36 | 2.46 | 0.95 | 8.23 |
| $S_2$-8 | 98.56 | 3.38 | 4 | 8 | 8.21×9.80 | 1.35×0.72 | 2.75×2.58 | 2.67 | 0.89 | 7.46 |
| $S_2$-9 | 158.46 | 4.78 | 5 | 42 | 8.63×10.20 | 1.35×0.68 | 3.08×2.59 | 4.39 | 1.16 | 10.95 |
| $S_2$-10 | 169.76 | 4.86 | 5 | 15 | 9.90×11.00 | 1.59×1.02 | 3.19×2.98 | 4.34 | 1.35 | 10.96 |
| *G. nelsonii* Fryx. | 80.00 | 3.45 | 4 | 7 | 5.21×6.18 | 1.10×1.32 | 3.56×2.87 | 2.18 | 0.85 | 17.70 |
| *G. herbaceum* L. | 200.00 | 4.96 | 7 | 15 | 12.86×13.98 | 1.2×0.25 | 2.31×2.06 | 3.53 | 1.48 | 5.60 |

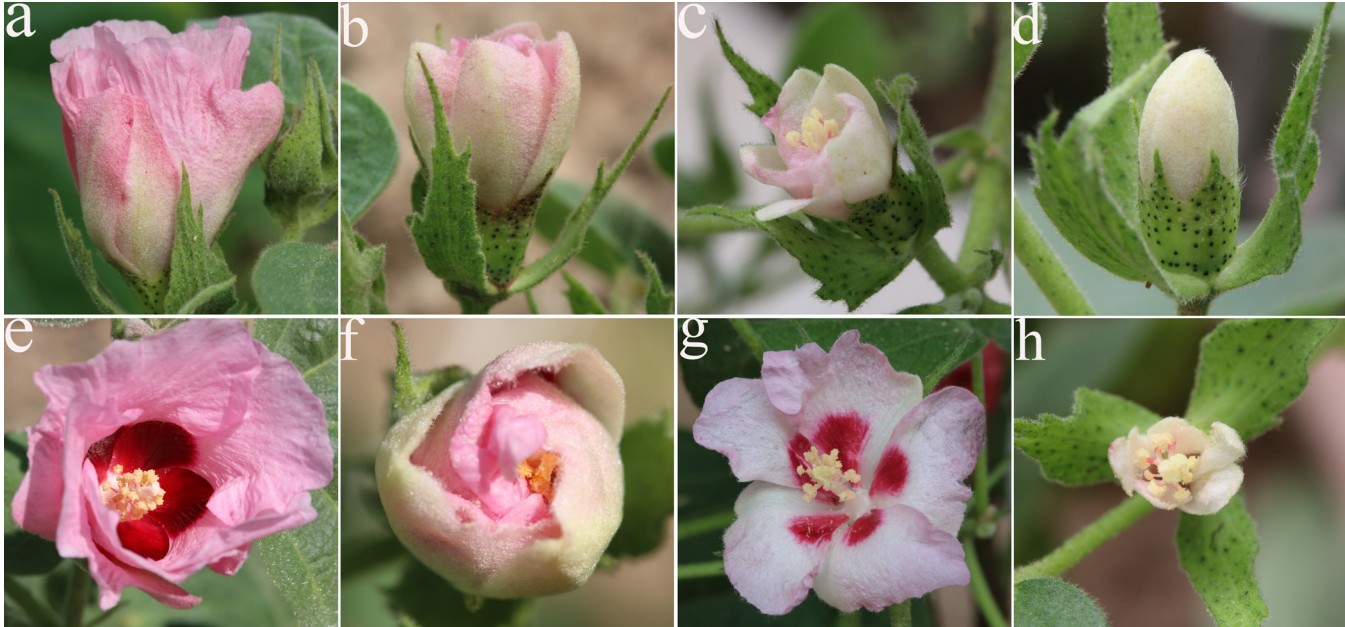

**Fig 4. Flowers of the tetraploid S2.** (a) Pink, cylindrical flower with pink corolla, (b) funnel-shaped flower with light pink corolla, (c) funnel-shaped flower with white and pink alternating corolla, (d) white, closed flower, (e) bell-shaped flower with red corolla and a clear rose-colored spot on the petal base, (f) semi-closed flower with pink corolla, (g) bell-shaped flower with pale pink corolla. (h) small flower with white corolla and no spot on the petal base.

It is also an ideal population for the screening of disease-resistant varieties using conventional hybridization and breeding methods.

## Determination of disease resistance in the tetraploid S2

For each plant, there were 108 nutrient pots, and each pot contained five seedlings. After the development of the first true leaf, cotton plants were inoculated with a fixed concentration of *Verticillium dahliae* Kleb. suspension (Fig 5A). After 20 days the percentage of infected plants in the control group was 80%. Results (Table 5) showed that when the inoculation was carried out indoors, the seedlings developed disease at the cotyledon stage. There were 456 plants from the susceptible strain cv. *Jimian11* that developed disease, with an incidence of 84.44% and disease index of 44.00 (Fig 5B). The resistant grades in S2 descendants of S1 were considered to be immune or highly resistant (Fig 5C). The incidence and disease indexes of *G*.

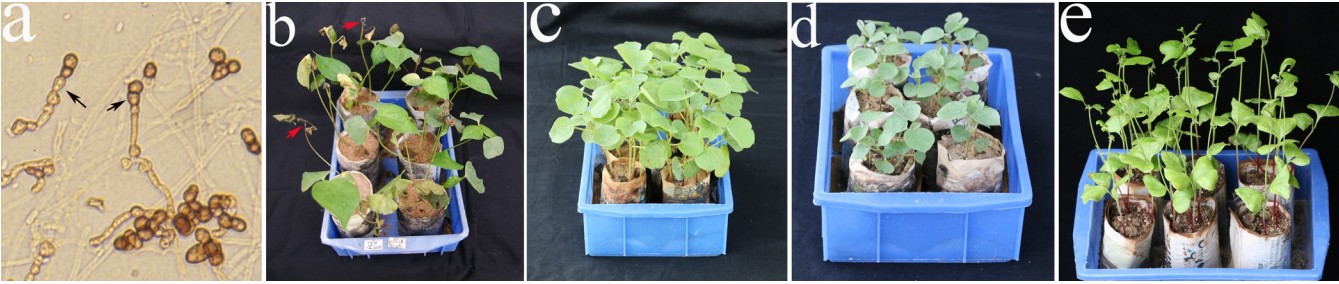

**Fig 5. Identification of resistance to *Verticillium dahliae* Kleb.** (a) The pathogen of the conidia is indicated with the black arrow. (b) Disease-susceptible upland cotton *Jimian11*, with yellowing, necrosis, and leaf abscission indicated by red arrows. (c) The resistant S1. (d) Resistant *G. nelsonii*. (e) Resistant *G. herbaceum*.

**Table 5. Identification of cotton Verticillium wilt resistance in the $S_2$ descendants of $S_1$ with their parental plants.**

| Cultivar | Disease index | Relative resistance index | Classification |
|---|---|---|---|
| *Jimian11* | 44.00±3.62 | 22.00±1.81 | S |
| $S_2$-1 | 0.00 | 0.00 | I |
| $S_2$-2 | 0.00 | 0.00 | I |
| $S_2$-3 | 0.28±0.68 | 0.14±0.34 | HR |
| $S_2$-4 | 0.00 | 0.00 | I |
| $S_2$-5 | 0.00 | 0.00 | I |
| $S_2$-6 | 0.56±0.12 | 0.28±0.06 | HR |
| $S_2$-7 | 0.00 | 0.00 | I |
| $S_2$-8 | 0.00 | 0.00 | I |
| $S_2$-9 | 0.16±2.86 | 0.8±1.43 | HR |
| $S_2$-10 | 0.00 | 0.00 | I |
| *G. nelsonii* Fryx. | 0.00 | 0.00 | I |
| *G. herbaceum* L. | 1.68±1.64 | 0.84±0.82 | HR |

*nelsonii* were zero, and the resistance was classified as immune (Fig 5D). The disease index of *G. herbaceum* was 1.68 (Fig 5E). No disease occurred in any of the plants inoculated with sterile water. From this, it was preliminarily determined that the tetraploid $S_2$ was germplasm material with potential disease resistance. In future studies, we will examine and validate the disease resistance in the $S_2$ in specific plots in the field, to provide reliable intermediate materials for the cultivation of disease-resistant *G. hirsutum* L.

## Description of the $S_1$ synthetic tetraploid hybrid based on morphological and fertility traits

From an analysis of the effective boll number and boll-forming rate (Table 6), we determined that the tetraploid $S_1$ showed some compatibility for hybridization with the TM-1. In 2018, the $S_1$ was hybridized with the TM-1 plants and 16 effective bolls were obtained, but they were small and deformed (S3A–S3D Fig). These grew into seven three-way $F_1$ plants, of which only two grew normally and were fertile (Fig 6A–6D). The remaining five plants did not grow normally; some of them grew until flowering stage and others stopped their growth having low fertility or sterility. This is an inevitable consequence of three-way species crossing between *G. hirsutum* L. × (*G. herbaceum* L. × *G. nelsonii* Fryx. $S_1$). Hence, the five three-way $F_1$ hybrids cannot be used for breeding.

The plants described above, that grew normally and were fertile, were transplanted to fields and named $F_1$-1 and $F_1$-2 (Fig 6C and 6D). Field morphological observations found that the $F_1$-1 plant retained significant wild cotton characteristics, which were mostly paternal traits (Fig 6F and 6G). The morphology of the $F_1$-2 was an intermediate type, having the most important traits from *G. hirsutum* L. (Fig 6E and 6H). The significant differences between the two plants were the corolla color, presence of spots, bract characteristics, trichomes, and boll fertility. The $F_1$-1 plant had a pink-colored corolla and red spots, bracts that were separate and not jointed (Fig 6G), and the stems, branches, leaves, and bolls had a large number of

**Table 6. Hybridization characteristics of TM-1× (*G. herbaceum* × *G. nelsonii* $S_1$).**

| No. of flowers pollinated | Effective boll number | Boll-forming rate (%) | No. of normal embryos | No. of three-way $F_1$ plants | No. of three-way $F_1$ low-fertility plants |
|---|---|---|---|---|---|
| 278 | 16 | 5.75 | 9 | 7 | 2 |

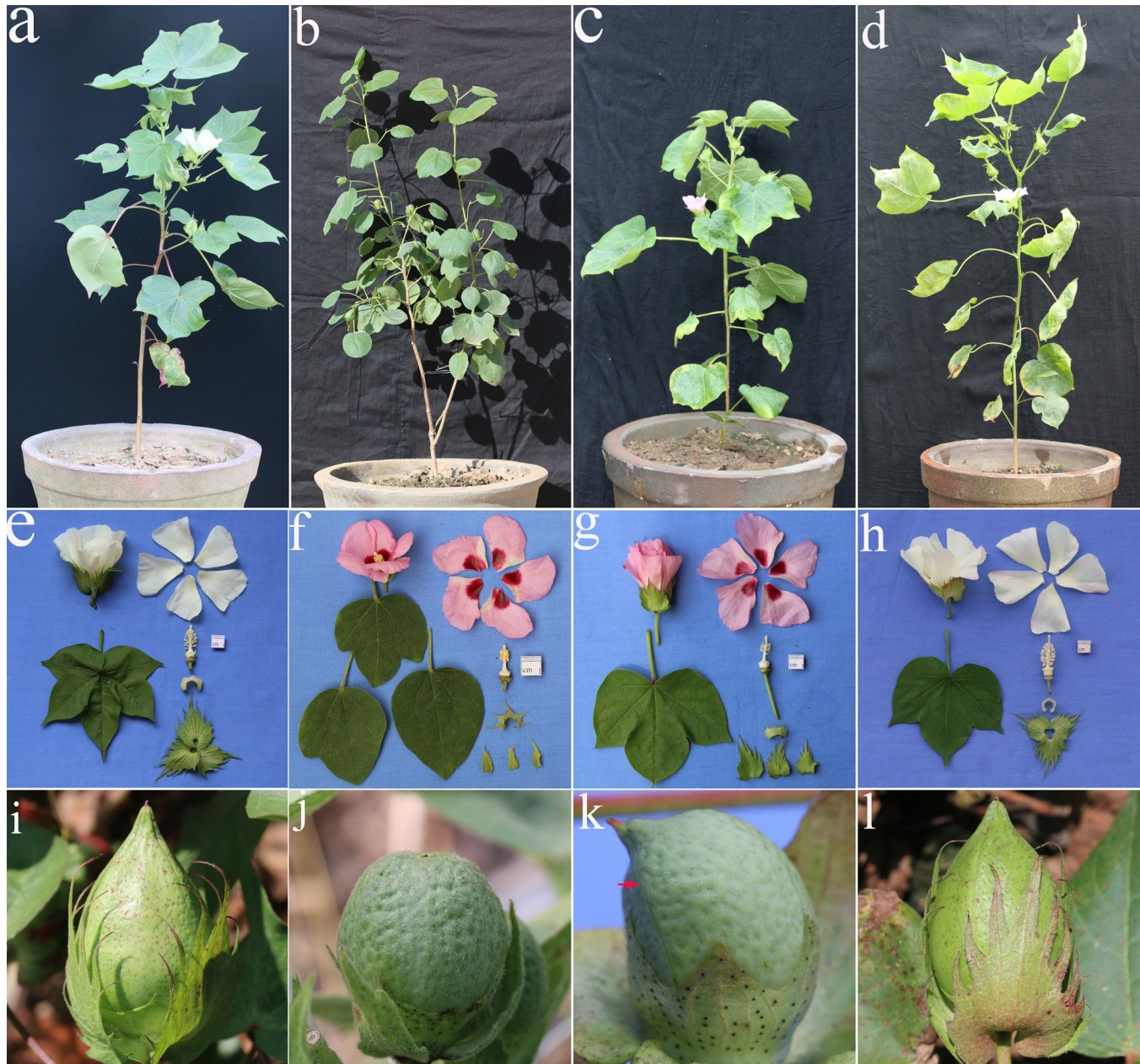

**Fig 6. The plants, leaves, flowers, corollas, basal patches, filaments, anthers, pollen, calyxes, bracts, and bolls about TM-1, S₁ and their offprings.** TM-1(a, e), $S_1$(b, f), $F_1$-1(c, g), $F_1$-2(d, h). (i) normal conical boll of cotton TM-1, (j) green pear-shaped boll of $S_1$, (k) downy green boll of $F_1$-1 that was deformed, indicated by the red arrow, and (l) normal conical boll of $F_1$-2.

trichomes. The boll number was low, and the bolls were deformed (Fig 6K). Furthermore, the embryos tested from the seeds were very small and infertile. The $F_1$-2 plant had a milky white corolla; no spots were present, and the bracts were jointed (Fig 6H). The surfaces of the stems, branches, leaves, and bolls were smooth and there were no trichomes. The bolls developed into a funnel shape and the seed embryos were large and fertile (Fig 6L).

The $S_1$ had some potential for breeding applications, but there was severe segregation in its progeny and its fertility was decreased. Notably, a limitation of our study is that we have only

preliminarily assessed the authenticity of the $F_1$ generation obtained from the crosses between the TM-1 and $S_1$ through morphological characterization. Further studies using cytology and molecular genetics techniques are required for understanding the nature and usefulness of this new germplasm.

## Discussion

Wild cotton varieties have undergone long-term natural selection to accumulate rich genetic diversity conferring adaptability to adverse conditions and resistance to environmental onslaughts and diseases. They provide a great treasury of traits that could be utilized to improve commercial cultivars [5]. Despite continuous improvements in yield and fiber quality of commercial cotton cultivars, disease resistance has weakened. Resistance breeding programs are now being conducted to transfer the disease resistance genes from wild cotton to cultivated cotton, and to re-diversify the genetic makeup of cultivated cotton.

The superior quality of cotton fiber and the disease resistance of wild cotton have been successfully transferred to cultivated cotton [29, 30]; However, diploid wild cotton and tetraploid cultivated cotton are biologically isolated. They are often genetically incompatible, forming a barrier to the full utilization of wild diploid cotton varieties [31–33]. We have already obtained an interspecific hybrid $F_1$ of *G. herbaceum* and *G. raimondii*. The $F_1$ plants are sterile in the field, leading to various methods being attempted to double their chromosome but without success [21]. In the present study, we successfully obtained an infertile interspecific $F_1$ hybrid and managed to overcome the genetic incompatibility barriers between two distantly related species. Furthermore, we doubled the chromosomal content of this $F_1$ hybrid through colchicine immersion, overcoming the considerable challenges of interspecific hybrid sterility, to successfully synthesize a new $S_1$ tetraploid cotton species.

Our results support the hypothesis that distant hybridization is an important driver of new mutations and species in plant evolution [34, 35]. Since the discovery that microsatellite markers are uniformly distributed on cotton chromosomes, SSR markers have been used to monitor the infiltration of heterologous chromosome fragments in the offspring of interspecific hybrids [36, 37]. In the present study, SSR detection revealed the presence of new specific bands in the hybrids that were not detected in the parents, indicating that new combinations or mutations of chromosomes are present in the interspecific progeny (Genomic A and genomic C). This result might be due to chromosomal rearrangements in this newly formed hybrid [38] which was consistent with those reported in previous studies on trispecific hybrids [20]. There is potential for genes from the new synthetic allotetraploids to fuse with favorable genes from *G. herbaceum* and *G. nelsonii*, including genes that encode good fiber quality, or resistance to diseases. The number of bolls per $S_1$ plant reached 50 (S2A Fig), an important characteristic for high yield breeding applications. We found that a variety of colored fibers developed in $S_2$ cotton bolls, including brown fibers in a 4-chambered boll and light green and brown cotton fibers in a 3-chambered boll (S2B and S2D Fig). The preliminary examination showed that the disease resistance of the tetraploid $S_2$ was better than *G. herbaceum* and the same level as *G. nelsonii*. Increased disease resistance also occurs in interspecific hybrids between *G. hirsutum* × *G. arboretum* [19]. This important agronomic characteristic provided a new and important resource for cotton disease resistance breeding programs.

At the same time, there is a need to control segregation in hybrids and establish new genetic isolates. We found that the $S_1$ progeny in the field showed a phenomenon of gradual decline in growth. The morphological traits of the $S_1$ plants were generally consistent, with normal growth, flowering, and fruiting. However, the $S_2$ plants showed dramatic segregation and resulted in different types of plant and corolla shapes, and huge differences in the flowering and fruiting characteristics. The growth of many plants in the $S_3$ to $S_5$ generations was

suppressed and only a few plants could flower. In severe cases, the plants could not flower and only a small number of plants could bear fruit. This shows that $S_1$ was genetically unstable germplasm and there is still a risk of natural segregation. The decline in growth potential for hybrid offspring is a common phenomenon in cotton distant hybridization [7, 15].

To determine the breeding potential of the $S_1$ population, the hybridization of $S_1$ with the TM-1 upland cotton germline yielded two different types of hybrid $F_1$, namely $F_1$-1 with red flowers and $F_1$-2 with white flowers, indicating that the different genotypes of parents lead to the diversity and complexity of their hybrid offspring in the process of genetic recombination. The $F_1$-1 plants died at the onset of anthesis, indicating that this plant type could not be utilized in crossbreeding. The sterility of $F_1$-1 hybrid plants was consistent with the results of interspecific hybrids of *G. hirsutum* L. and *G. tomentosum* Nutt. [39], which may be caused by the incoordination of physiological processes in the hybrid offspring. The $F_1$-2 plants grew normally until flowering and fruiting, but their seeds could not germinate normally. This phenomenon of unsuccessful germination of $F_1$-2 seeds and root tip wilting had previously been observed during the early generations of the amphiploid *G. arboretum* L. hybridized with *G. bickii* Prokh. [12]. Additional research is required to overcome the key difficulties in promoting seed germination in each generation. The variation in hybrid phenotypes may be related to gene overlap, new genetic variation, chromosome rearrangement, or a variety of other factors [40, 41].

The tetraploid $S_1$ is a new germplasm obtained by chromosome doubling of the diploid $F_1$ hybrid and is a polyploidy event in cotton. This restored fertility to sterile diploid $F_1$ hybrids, and the tetraploid $S_1$ generation obtained showed segregation, which increases the phenotypic mutations in the species, enabling breeders to select for required traits. After the polyploid was generated, not only did its stress resistance not decrease, but biotic stress resistance was maintained. In addition, the fiber quality of the tetraploid $S_2$ exceeded that of both parents. The $S_2$ generation had high potential for breeding owing to its boll-forming properties, diversity of fiber colors, and fiber quality. These traits are also of great value for breeding programs. From this we can see that diploid $F_1$ hybrid polyploidy has important significance for cotton evolution; as the genome structure changed, the species genetic diversity was enriched, and species adaptability and fiber quality was improved. Thus far, we have only performed preliminary characterizations of the newly synthesized allotetraploid cotton $S_1$. It was suggested a more detailed in-depth and comprehensive research to extract the valuable information from this new cotton tetraploid $S_1$, so as to provide a more accurate theoretical basis for cotton breeding and genetic evolution.

## Supporting information

**S1 Fig. Growth and formation of the tetraploid $S_1$.** (a) Tetraploid plants with a large pink flower accompanied by yellow pollen. (b) The growth process of the oval boll. (c) Four seeds with brown fibers.
(TIF)

**S2 Fig. Fruits and Fibers of the $S_1$.** (a) Multiple bolls of $S_1$ on multiple fruit branches with the sunken navel of the boll tip showing an ergonomic shape with strong bolls. (b) Four-chambered boll with brown fibers. (c) The brown fibers with fiber length of 13 mm. (d) Three-chambered boll with light green and brown cotton fibers.
(TIF)

**S3 Fig. Cotton bolls of the TM-1 and $F_1$ hybrid (TM-1 × $S_1$).** (a)The normal conical boll of TM-1 shown by the red arrow. The abnormally conical boll of $F_1$ on the right (b) and on the left (c). (d) The deformed hybrid boll of $F_1$ indicated by yellow arrows.
(TIF)

## Acknowledgments

We are grateful to Dr Kunbo Wang and Dr Fang Liu of the Cotton Research Institute of Chinese Academy of Agricultural Sciences for kindly providing us with the pollen of *G. nelsonii* Fryx. to use at the National Wild Cotton Nursery, Sanya, China.

## Author Contributions

**Conceptualization:** Xiaomin Yin, Di Chen.

**Data curation:** Lixia Wang.

**Funding acquisition:** Xiaomin Yin.

**Investigation:** Yu Ge.

**Methodology:** Xiaomin Yin, Di Chen.

**Project administration:** Di Chen.

**Software:** Lixia Wang, Yu Ge.

**Writing – original draft:** Xiaomin Yin, Rulin Zhan.

**Writing – review & editing:** Xiaomin Yin, Rulin Zhan, Yingdui He, Shun Song.

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
