## [Decision Letter · Decision Letter 0]

1 Apr 2020

PONE-D-20-02970

Morphological and genetic description of a novel synthetic allotetraploid of Gossypium herbaceum L. and Gossypium nelsonii Fryx.

PLOS ONE

Dear Mrs. Yin,

Thank you for submitting your manuscript to PLOS ONE. After careful consideration, we feel that it has merit but does not fully meet PLOS ONE’s publication criteria as it currently stands. Therefore, we invite you to submit a revised version of the manuscript that addresses the points raised during the review process.

We would appreciate receiving your revised manuscript by May 16 2020 11:59PM. To enhance the reproducibility of your results, we recommend that if applicable you deposit your laboratory protocols in protocols.io, where a protocol can be assigned its own identifier (DOI) such that it can be cited independently in the future. For instructions see: http://journals.plos.org/plosone/s/submission-guidelines#loc-laboratory-protocols

We look forward to receiving your revised manuscript.

Kind regards,

Paulo Eduardo Teodoro, Dr.

Academic Editor

PLOS ONE

Journal Requirements:

2. In your Methods section, please provide additional details regarding the cotton varieties used in your study and ensure you have described the source. For more information regarding PLOS' policy on materials sharing and reporting, see https://journals.plos.org/plosone/s/materials-and-software-sharing#loc-sharing-materials.

Additional Editor Comments (if provided):

Dear authors, your manuscript was reviewed by three experts and all of them pointed out important changes to be made in the manuscript. I ask that you respond to all comments made in a letter-by-point response letter and mark corrections in red in the text. The English in this manuscript is also not acceptable. Please look for a specialized company (I suggest Editage or AJE) for English review and send the review certificate as supplementary material. If the manuscript is not in proper English, it will not be returned to the reviewers for the next round of evaluation and this will delay your evaluation process.

Reviewers' comments:

Reviewer's Responses to Questions

**Comments to the Author**

1. Is the manuscript technically sound, and do the data support the conclusions?

Reviewer #1: Yes

Reviewer #2: Yes

Reviewer #3: Yes

2. Has the statistical analysis been performed appropriately and rigorously? 

Reviewer #1: Yes

Reviewer #2: Yes

Reviewer #3: No

3. Have the authors made all data underlying the findings in their manuscript fully available?

Reviewer #1: Yes

Reviewer #2: Yes

Reviewer #3: Yes

4. Is the manuscript presented in an intelligible fashion and written in standard English?

Reviewer #1: Yes

Reviewer #2: Yes

Reviewer #3: No

5. Review Comments to the Author

Reviewer #1: The manuscript “Morphological and genetic description of a novel synthetic allotetraploid of Gossypium herbaceum L. and Gossypium nelsonii Fryx” created a new germplasm of cotton through remote hybridisation for a resistant source of wild cotton. G. herbaceum L. crossed with G. nelsonii Fryx. to produce an interspecies F1 hybrid cotton. Colchicine was used to double its chromosomal content, leading to the successful generation of a synthetic S1 tetraploid. The newly created S1 tetraploid showed morphological variations in its flowers, cotton bolls, and fibres and was partially resistant to Verticillium dahliae Kleb . The S1 tetraploid was hybridised with upland cotton to produce two types of cotton with different characteristics. This newly synthesised tetraploid cotton species is a valuable resource for cotton resistance breeding and for the enrichment of cotton genetic resources. The results are informative, but there are some major concerns need to be addressed.

1. The description of phenotypic morphological analysis of the diploid F1 hybrid and the synthetic S1 tetraploid is poorly organized. The author's investigation of morphological in materials and methods has not been described in detail. Even with reference to the methods of previous reference, no specific survey data and criteria are given. In the survey results, the key traits such as the germinating stage and the flowering stage are unclear. Moreover, wild cotton is mostly perennial wild material, and its germinating stage and flowering stage belongs to agronomic traits. It is not suitable for investigation and comparative analysis of wild materials.

2. Statistical analysis of leaf shape, plant height, boll shape and other traits is not appropriate. Because the leaf shape will change according to the different growth stages of the plant, the plant height will vary greatly due to changes in the growth environment, and the boll shape will also vary in different positions of the plant. The number of bolls per plant is also strongly related to nutritional status and planting density, so the author's description of these traits seems meaningless.

3. When statistically analysis the differences in phenotypic traits, it is necessary to distinguish between qualitative and quantitative traits. For F1, in the case of a limited single plant, it cannot be compared with a tetraploid with a larger number of offspring. As for the results of the investigation against Verticillium wilt, as it is a quantitative trait, it is recommended to compare and analyze the resistance differences of different isolated plants of S1.

4. For the identification and analysis of F1 true or false hybrids, when doing cytological analysis, the chromosomes of both parents and the chromosome of F1 should be observed.

5. When conducting a flow cytometer test, both parents, F1, S1, and the TM-1 materials, should be tested and investigated to increase the scientificity and credibility of the test.

6. At present, according to the taxonomic standards of the cotton genus, it has been recognized that Gossypium nelsonii Fryxell has been classified as the G3 species. So the newly synthesized double cotton germplasm should be A1A1G3G3.

7. According to the cotton genus classification and distribution research, 53-54 cotton species have been discovered and named, instead of 55.

Reviewer #2: The manuscript has potential for publication, i the authors have achieved good results, the discussion needs to be improved, with mention to other works, in the discussion it is necessary to mention other results found in the difference of ploidy, the use of colchicine, as well as resistance to Verticillium dahliae, which is one of the main objectives, however, the work has all merit for publication.

Reviewer #3: WHAT THE AUTHORS NEED TO DO FOR IMPROVE THEIR PAPER

• I THEORIZED THAT YOU MUST FOLLOW A LANGUGE EDITING FIRSTLY AND AFTERWARDS WE COULD CAPABLE TO FOLLOW DETAILED CORRECTIONS AND SUGGESTIONS.

• I AM SORRY BUT THERE MAINY TERMS AND SENTENCES THAT MUST BE REWRITE USING BETTER TERMINOLOGY.

ALL THE SENTENCES, TERMS AND WORDS UNDERLINED INTO THE TEXT BY YELLOW COLOUR SHOULD BE CHANGED.

• PLEASE FOLLOW A LANGUAGE EDITIND FROM A PERSON WHO HAS ENGLISH AS NATIVE LANGUAGE BECAUSE YOU LOOSE MUCH OF THE VALUE OF YOUR WORK .

PLEASE SEE CAREFULLY OUR ADVICES INTO THE FILE COMMENTS & SUGGESTIONS (1)

6. PLOS authors have the option to publish the peer review history of their article (what does this mean?). If published, this will include your full peer review and any attached files.

Reviewer #1: No

Reviewer #2: No

Reviewer #3: No

---

## [Author Response · Author response to Decision Letter 0]

29 May 2020

1. The description of phenotypic morphological analysis of the diploid F1 hybrid and the synthetic S1 tetraploid is poorly organized. The author's investigation of morphological in materials and methods has not been described in detail. Even with reference to the methods of previous reference, no specific survey data and criteria are given. In the survey results, the key traits such as the germinating stage and the flowering stage are unclear. Moreover, wild cotton is mostly perennial wild material, and its germinating stage and flowering stage belongs to agronomic traits. It is not suitable for investigation and comparative analysis of wild materials.

__Answer：We have added the key traits such as germination and flowering of new germplasm cotton.The F1 hybrid were sown in a pot and all germinated after 5 days. After the first true leaves were level, the seedlings were transplanted to the field. Floral buds appeared after 43-45 days and flowering occurred after 68-70 days. The S1 were sown in a pot and all germinated after 5 days. After the first true leaves were level, the seedlings were transplanted to the field. Floral buds appeared after 45-50 days and flowering occurred after 70-75 days. Boll opening occurred after 107-115 days.

The new cotton germplasm was created by using wild cotton through distant hybridization and is the descendant of wild cotton.When investigating and analyzing it, it involves genetic problems.To explain the genetic problems of new germplasm, the source needs to be traced back. Therefore, it is necessary to list wild cotton parents as a reference in the study in order to analyze and explain.The reviewer mentioned that wild cotton is mostly perennial wild material, this statement is correct.

In fact, all cotton is a perennial plant in tropical regions, including wild cotton and cultivated cotton (such as G. hirsutum L. G. barbadense L.G. herbaceum L.G. arboreum L. ). The morphological traits of various cotton species grown in the same environment difference.

2. Statistical analysis of leaf shape, plant height, boll shape and other traits is not appropriate. Because the leaf shape will change according to the different growth stages of the plant, the plant height will vary greatly due to changes in the growth environment, and the boll shape will also vary in different positions of the plant. The number of bolls per plant is also strongly related to nutritional status and planting density, so the author's description of these traits seems meaningless.

__Answer：Plant morphology is a characteristic expression of a species, mainly including plant morphology traits such as plant type, stem, leaf, flower and cotton boll.Regarding the color of the floral organs of the plant, the presence or absence of the floral basal spot, the characteristics of the bracts, the amount of plant hairs, and the boll fertility is a trait that can be stably inherited and does not change with the external environment.In a limited time, these descriptions of the new germplasm we created, please dear editors and review experts to understand.We describe morphology and only want to show the new cotton germplasm as much as possible. If we talk about its significance, the birth event of new cotton germplasm is meaningful in itself.

3. When statistically analysis the differences in phenotypic traits, it is necessary to distinguish between qualitative and quantitative traits. For F1, in the case of a limited single plant, it cannot be compared with a tetraploid with a larger number of offspring. As for the results of the investigation against Verticillium wilt, as it is a quantitative trait, it is recommended to compare and analyze the resistance differences of different isolated plants of S1.

__Answer：Due to the low number of diploid F1 hybrids and tetraploid S1 plants, these two materials cannot be used for cotton Verticillium wilt resistance experiments.In fact, in this study, we used the descendant material S2 of S1 to carry out disease resistance identification of seedlings in indoor nutrition bowls（Fig.5）, because S2 material is not only a large number, but also an ideal separation group with rich morphological diversity .In the next step, we will plant S2 in a specific field nursery, and analyze and compare the differences in disease resistance of different S2 isolated plants.

4. For the identification and analysis of F1 true or false hybrids, when doing cytological analysis, the chromosomes of both parents and the chromosome of F1 should be observed.

__Answer：We have shown the cytological observations of both parents, F1 and S1 in the article.

5. When conducting a flow cytometer test, both parents, F1, S1, and the TM-1 materials, should be tested and investigated to increase the scientificity and credibility of the test.

__Answer：We have shown the results of flow cytometer test of female parent, F1, S1 and TM-1 in this article.

6. At present, according to the taxonomic standards of the cotton genus, it has been recognized that Gossypium nelsonii Fryxell has been classified as the G3 species. So the newly synthesized double cotton germplasm should be A1A1G3G3.

__Answer：We have changed to A1A1G3G3 in the article.

7. According to the cotton genus classification and distribution research, 53-54 cotton species have been discovered and named, instead of 55.

__Answer：We have changed to 54 kinds in the article.

Reviewer #2:

 The manuscript has potential for publication, i the authors have achieved good results, the discussion needs to be improved, with mention to other works, in the discussion it is necessary to mention other results found in the difference of ploidy, the use of colchicine, as well as resistance to Verticillium dahliae, which is one of the main objectives, however, the work has all merit for publication.

__Answer：We have added relevant content to the discussion of the article. The tetraploid S1 is a new germplasm obtained by chromosome doubling of the diploid F1 hybrid and is a polyploidy event in cotton. This not only restores fertility to sterile diploid F1 hybrids and the tetraploid S1 generation obtained showed segregation, which increases the phenotypic mutations in the species, enabling breeders to select for required traits. After the polyploid was generated, its stress resistance not only did not decrease but biotic stress resistance was maintained. For example, the preliminary examination showed that the disease resistance of the tetraploid S2 was 1 grade better than the female parent (G. herbaceum) and the same level as the male parent (G. nelsonii), and its disease resistance inheritance was closer to the male parent. In addition, the fiber quality of the tetraploid S2 exceeded that of both parents. From this we can see that diploid F1 hybrid polyploidy has important significance for cotton evolution, as the genome structure changed and the species genetic diversity enriched as well as the species adaptability and fiber quality were improved.

Reviewer #3: 

WHAT THE AUTHORS NEED TO DO FOR IMPROVE THEIR PAPER• I THEORIZED THAT YOU MUST FOLLOW A LANGUGE EDITING FIRSTLY AND AFTERWARDS WE COULD CAPABLE TO FOLLOW DETAILED CORRECTIONS AND SUGGESTIONS.• I AM SORRY BUT THERE MAINY TERMS AND SENTENCES THAT MUST BE REWRITE USING BETTER TERMINOLOGY.ALL THE SENTENCES, TERMS AND WORDS UNDERLINED INTO THE TEXT BY YELLOW COLOUR SHOULD BE CHANGED.• PLEASE FOLLOW A LANGUAGE EDITIND FROM A PERSON WHO HAS ENGLISH AS NATIVE LANGUAGE BECAUSE YOU LOOSE MUCH OF THE VALUE OF YOUR WORK .

PLEASE SEE CAREFULLY OUR ADVICES INTO THE FILE COMMENTS & SUGGESTIONS (1)

__Answer：We have made changes to the questions raised by the review experts.

---

## [Decision Letter · Decision Letter 1]

26 Jun 2020

PONE-D-20-02970R1

Morphological and genetic description of a novel synthetic allotetraploid of Gossypium herbaceum L. and Gossypium nelsonii Fryx.

PLOS ONE

Dear Dr. Chen,

Thank you for submitting your manuscript to PLOS ONE. After careful consideration, we feel that it has merit but does not fully meet PLOS ONE’s publication criteria as it currently stands. Therefore, we invite you to submit a revised version of the manuscript that addresses the points raised during the review process.

Dear authors, your manuscript has been revised again by the same reviewers of round 1.

One of them refused the article, as several suggested changes were not made. I agree with this reviewer.

The other reviewer made a detailed review of the text and pointed out several excerpts that must be modified.

The English in this manuscript remains poor and very difficult to read. I strongly suggest that the authors do an English review for the next round of evaluation. I suggest EDITAGE or AJE. Please attach the English revision certificate as a supplementary document. Without this review, it will be difficult to proceed with the evaluation of this manuscript.

We look forward to receiving your revised manuscript.

Kind regards,

Paulo Eduardo Teodoro, Dr.

Academic Editor

PLOS ONE

Additional Editor Comments (if provided):

Dear authors, your manuscript has been revised again by the same reviewers of round 1.

One of them refused the article, as several suggested changes were not made. I agree with this reviewer.

The other reviewer made a detailed review of the text and pointed out several excerpts that must be modified.

The English in this manuscript remains poor and very difficult to read. I strongly suggest that the authors do an English review for the next round of evaluation. I suggest EDITAGE or AJE. Please attach the English revision certificate as a supplementary document. Without this review, it will be difficult to proceed with the evaluation of this manuscript.

Reviewers' comments:

Reviewer's Responses to Questions

**Comments to the Author**

1. If the authors have adequately addressed your comments raised in a previous round of review and you feel that this manuscript is now acceptable for publication, you may indicate that here to bypass the “Comments to the Author” section, enter your conflict of interest statement in the “Confidential to Editor” section, and submit your "Accept" recommendation.

Reviewer #1: (No Response)

Reviewer #2: All comments have been addressed

Reviewer #3: All comments have been addressed

2. Is the manuscript technically sound, and do the data support the conclusions?

Reviewer #1: No

Reviewer #2: Yes

Reviewer #3: Partly

3. Has the statistical analysis been performed appropriately and rigorously? 

Reviewer #1: No

Reviewer #2: Yes

Reviewer #3: Yes

4. Have the authors made all data underlying the findings in their manuscript fully available?

Reviewer #1: No

Reviewer #2: Yes

Reviewer #3: No

5. Is the manuscript presented in an intelligible fashion and written in standard English?

Reviewer #1: No

Reviewer #2: Yes

Reviewer #3: No

6. Review Comments to the Author

Reviewer #1: The author's concept of the distinction between wild cotton and cultivated cotton is vague. The content of the survey on qualitative and quantitative traits is unclear. The author is confused about the genomic properties of Gossypium. No substantial changes were given to all amendment proposals.

Reviewer #2: (No Response)

Reviewer #3: GENERAL COMMENT

Although the authors incorporated the suggestions from the reviewers, I feel that this paper must be improved, by English Editing and a careful extension of discussion with mention to other research works and results found in international references.

I propose some differentiation in the subtitles and in titles of the tables (please find STIKCY NOTES in the manuscript).

(e.g p.p. 173 A title : Root tip chromosome observations changed as KARYOTYPE ANALYSIS)

All new suggestions and corrections are indicated by Highlight Notes in the Manuscript (please change all the proposed words or terms or terminology as proposed in the NOTE BOX).

NOTICE : Micronaire value up to 6, is not a desirable trait for fiber quality.

Please not confuse the reader.

THE RESULTS FROM THIS RESEARCH ARE VERY IMPORTANT.

The researchers offer lot of work onto these experiments but is important to clarify the useful data and give the most significant conclusions in a clear way.

PLEASE WRITE THE CONCLUSIONS

7. PLOS authors have the option to publish the peer review history of their article (what does this mean?). If published, this will include your full peer review and any attached files.

Reviewer #1: No

Reviewer #2: No

Reviewer #3: No

---

## [Author Response · Author response to Decision Letter 1]

20 Sep 2020

The concerns raised by the reviewers have been addressed in our revised manuscript. We carefully checked the language of the manuscript according to the comments of Reviewer 3 and the sticky notes in the pdf file to ensure a high level of readability in English. Please see below for our point-by-point response.

---

## [Decision Letter · Decision Letter 2]

28 Oct 2020

PONE-D-20-02970R2

Morphological description of a novel synthetic allotetraploid  (A1A1G3G3) of Gossypium herbaceum L. and G. nelsonii Fryx. suitable for disease-resistant breeding applications

PLOS ONE

Dear Dr. Chen,

Thank you for submitting your manuscript to PLOS ONE. After careful consideration, we feel that it has merit but does not fully meet PLOS ONE’s publication criteria as it currently stands. Therefore, we invite you to submit a revised version of the manuscript that addresses the points raised during the review process.

We look forward to receiving your revised manuscript.

Kind regards,

Paulo Eduardo Teodoro, Dr.

Academic Editor

PLOS ONE

Additional Editor Comments (if provided):

Dear authors, one of the reviewers also pointed out the need for Minor Revision in his manuscript.

Therefore, I ask you to carefully review the manuscript according to this reviewer's suggestions.

Reviewers' comments:

Reviewer's Responses to Questions

**Comments to the Author**

1. If the authors have adequately addressed your comments raised in a previous round of review and you feel that this manuscript is now acceptable for publication, you may indicate that here to bypass the “Comments to the Author” section, enter your conflict of interest statement in the “Confidential to Editor” section, and submit your "Accept" recommendation.

Reviewer #3: All comments have been addressed

2. Is the manuscript technically sound, and do the data support the conclusions?

Reviewer #3: Yes

3. Has the statistical analysis been performed appropriately and rigorously? 

Reviewer #3: Yes

4. Have the authors made all data underlying the findings in their manuscript fully available?

Reviewer #3: Yes

5. Is the manuscript presented in an intelligible fashion and written in standard English?

Reviewer #3: Yes

6. Review Comments to the Author

Reviewer #3: ALL CORRECTIONS AND SUGGESTIONS INCLUDED IN THE ATTACHED FILES (I) (Full Text with Suggestions )

AND (II) (Reviewer Comments)

7. PLOS authors have the option to publish the peer review history of their article (what does this mean?). If published, this will include your full peer review and any attached files.

Reviewer #3: No

---

## [Author Response · Author response to Decision Letter 2]

3 Nov 2020

We carefully checked the language of the manuscript according to the comments of Reviewer 3 and the sticky notes in the pdf file to ensure the rationality of the article structure.The concerns raised by the reviewers have been addressed and added the related references in our revised manuscript. In the process of revising the article, thank you for your patient review again and again in line with the realistic attitude, respecting the originality of science. Let us deeply feel the precision of your words in place, and increase our knowledge. These valuable experiences will be remembered for us forever.

---

## [Editor Report · Decision Letter 3]

6 Nov 2020

Morphological description of a novel synthetic allotetraploid  (A1A1G3G3) of Gossypium herbaceum L. and G. nelsonii Fryx. suitable for disease-resistant breeding applications

PONE-D-20-02970R3

Dear Dr. Chen,

We’re pleased to inform you that your manuscript has been judged scientifically suitable for publication and will be formally accepted for publication once it meets all outstanding technical requirements.

Kind regards,

Paulo Eduardo Teodoro, Dr.

Academic Editor

PLOS ONE
---

## [Editor Report · Acceptance letter]

18 Nov 2020

PONE-D-20-02970R3 

Morphological description of a novel synthetic allotetraploid(A_1_A_1_G_3_G_3_) of *Gossypium herbaceum* L.and *G.nelsonii* Fryx. suitable for disease-resistant breeding applications 

Dear Dr. Chen:

I'm pleased to inform you that your manuscript has been deemed suitable for publication in PLOS ONE. Congratulations! Your manuscript is now with our production department. 

Kind regards, 

on behalf of

Professor Paulo Eduardo Teodoro 

Academic Editor

PLOS ONE